# Meta-Neighborhoods

**Siyuan Shan**
Department of Computer Science
University of North Carolina at Chapel Hill
siyuanshan@cs.unc.edu

**Yang Li**
Department of Computer Science
University of North Carolina at Chapel Hill
yangli95@cs.unc.edu

**Junier B. Oliva**
Department of Computer Science
University of North Carolina at Chapel Hill
joliva@cs.unc.edu

## Abstract

Making an adaptive prediction based on one's input is an important ability for general artificial intelligence. In this work, we step forward in this direction and propose a semi-parametric method, Meta-Neighborhoods, where predictions are made adaptively to the neighborhood of the input. We show that Meta-Neighborhoods is a generalization of $k$-nearest-neighbors. Due to the simpler manifold structure around a local neighborhood, Meta-Neighborhoods represent the predictive distribution $p(y \mid x)$ more accurately. To reduce memory and computation overhead, we propose induced neighborhoods that summarize the training data into a much smaller dictionary. A meta-learning based training mechanism is then exploited to jointly learn the induced neighborhoods and the model. Extensive studies demonstrate the superiority of our method.[1]

## 1   Introduction

Discriminative machine learning models typically learn the predictive distribution $p(y \mid x)$. There are two paradigms to build a model, parametric methods and non-parametric methods [12]. Parametric methods assume that a set of *fixed* parameters $\theta$ dominates the predictive distribution, i.e., $p(y \mid x; \theta)$. The training process estimates $\theta$ and then discard the training data completely, as the learned parameters $\theta$ are responsible for the following prediction. This paradigm has proven effective, however, it leaves the entire burden on learning a complex predictive distribution over potentially large support. Non-parametric models differ in that the number of parameters scales with data. They typically reuse the training data during the testing phase to make predictions. For instance, the well-known $k$-nearest neighbor (KNN) estimator often achieves surprisingly good results by leveraging neighbors from the training data, which reduces the problem to a much simpler local-manifold. Despite its flexibility, non-parametric methods are required to store the training data and traverse them during testing, which may impose significant memory and computation overhead for large training sets.

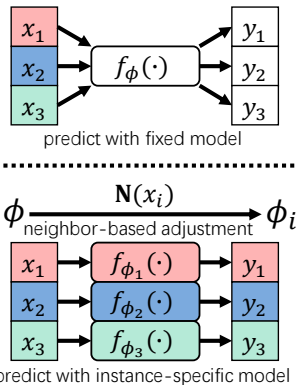

Figure 1: Top: traditional parametric models. Bottom: our per-instance adapted model.

In this work, we combine the merits of both paradigms and propose a semi-parametric method called Meta-Neighborhoods. The main body of Meta-Neighborhoods is a parametric neural network, but we adapt its parameters to a local neighborhood in a non-parametric scheme. The prediction is made on the local manifold by the adapted model. Fig. 1 illustrates the difference between traditional parametric models and the proposed model. Inspired by the success of inducing point methods from sparse Gaussian process literature [34, 38] to alleviate the storage burden and reduce time complexity, we learn induced neighborhoods, which summarize the training data into a much smaller dictionary. The induced neighborhoods and the neural network parameters are learned jointly.

Our model is also closely related to locally linear embeddings [30], which reconstructs the non-linear manifold with locally linear approximation around each neighborhood. In our method, we adapt an initial model (not necessarily linear) to local neighborhoods. Since the local manifold is much simpler, we expect the adapted model can better capture the predictive distribution. Overall, it learns a better discriminative model on the entire support.

Our method imposes challenges of adapting the initial model since the local neighborhoods usually do not contain enough training instances to independently adapt the model, and the induced neighborhoods contain even fewer instances. Inspired by the few-shot and meta-learning literature [6], we propose a meta-learning based training mechanism, where we learn an initial model so that it adapts to the local neighborhood after only several finetuning steps over a few instances inside the neighborhood.

The prediction process of our model remains flexible by following a non-parametric scheme. An input $x$ is first paired with its neighbors by querying the induced dictionary. The initial model is adapted to its neighborhood by finetuning several steps on the neighbors. We then predict the target $y$ using the adapted model.

Our contributions are as follows:

- We combine parametric and non-parametric methods in a meta-learning framework.
- We propose Meta-Neighborhoods to jointly learn the induced neighborhoods and an adaptive initial model, which can adapt its parameters to a local neighborhood according to the input through both finetuning and a proposed instance-wise modulation scheme, iFiLM.
- Extensive empirical studies demonstrate the superiority of Meta-Neighborhoods for both regression and classification tasks.
- We empirically find the induced neighbors are semantically meaningful: they represent informative boundary cases on both realistic and toy datasets; they also capture sub-category concepts even though such information is not given during training.

## 2 Method

**Problem Formulation** Given a training set $\mathcal{D} = \{(x_i, y_i)\}_{i=1}^N$ with $N$ input-target pairs, we learn a discriminative model $f_\phi(x)$ and a dictionary $M = \{(k_j, v_j)\}_{j=1}^S$ jointly from $\mathcal{D}$. The learned dictionary stores the neighbors induced from the training set, where $S$ is the number of induced neighbors. Just like the real training set $\mathcal{D}$, the dictionary stores input-target pairs as key-value pairs $(k_j, v_j)$, where both the keys and the values are learned end-to-end with the model parameters $\phi$ in a meta-learning framework. For classification tasks, $v_j$ is a vector representing the class probabilities while for regression tasks $v_j$ is the regression target. In the following text, we will use the terms "induced neighbors" and "learnable neighbors" interchangeably. We defer the exact training mechanism to Section 2.2.

### 2.1 Predict with Induced Neighborhoods

In this section, we assume access to the learned neighborhoods in $M$ and the learned model $f_\phi$. Different from the conventional parametric setting, where the learned model is employed directly to predict the target, we adapt the model according to the neighborhoods retrieved from $M$ and the adapted model is the one responsible for making predictions. Specifically, for a test data $x_i$, relevant entries in $M$ are retrieved in a soft-attention manner by comparing $x_i$ to $k_j$ via a attention function $\omega$.

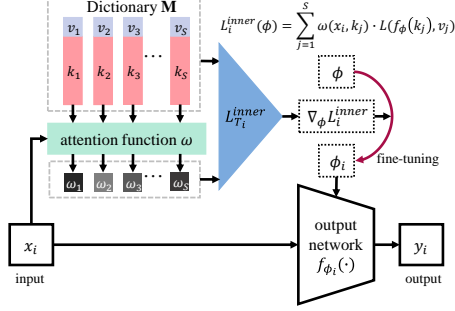

Figure 2: Model overview.

**Algorithm 1** META-NEIGHBORHOODS: TRAINING PHASE

**Require:** $\omega$: similarity metric, $\eta$: outer loop learning rate
1: Initialize $\theta, \phi, \alpha, M = \{(k_j, v_j)\}_{j=1}^S$
2: **while** not done **do**
3:     Sample a batch of training data $\{(x_i, y_i)\}_{i=1}^B$
4:     **for all** $(x_i, y_i)$ in current batch **do**
5:         Compute the feature vector $z_i = \mu_\theta(x_i)$
6:         Compute $\mathcal{L}_i^{\text{inner}}(\phi) = \sum_{j=1}^S \omega(z_i, k_j) L(f_\phi(k_j), v_j)$
7:         Finetune $\phi$: $\phi_i = \phi - \alpha \nabla_\phi \mathcal{L}_i^{\text{inner}}(\phi)$
8:     **end for**
9:     Compute $\mathcal{L}^{\text{meta}}(\phi, \theta, M, \alpha) = \frac{1}{B} \sum_{i=1}^B L(f_{\phi_i}(\mu_\theta(x_i)), y_i)$
10:    Update model parameters $\Theta = \{\theta, \phi, M, \alpha\}$
       using gradient descent as $\Theta \leftarrow \Theta - \eta \nabla_\Theta \mathcal{L}^{\text{meta}}$
11: **end while**

The retrieved entries are then utilized to finetune $f_\phi$ following

$$\phi_i \leftarrow \phi - \alpha \nabla_\phi \sum_{j=1}^S \omega(x_i, k_j) L(f_\phi(k_j), v_j), \tag{1}$$

where $\alpha$ is the finetuning step size. Note we weight the loss terms of all dictionary entries by their similarities to the input test data $x_i$. The intuition is that nearby neighbors play a more important role in placing $x_i$ into the correct local manifold. Since the model $f_\phi$ is specially trained in a meta-learning scheme, it adapts to the local neighborhoods after only a few finetuning steps.

To better understand our method, we draw connections to other well-known techniques. The above prediction process is similar to a one-step EM algorithm. Specifically, the dictionary querying step is an analogy to the Expectation step, which determines the latent variables (in our case, the neighborhood assignment). And the finetuning step is similar to the Maximization step, which maximizes the expected log-likelihood. We can also view this process from a Bayesian perspective, where the initial parameter $\phi$ is an empirical prior estimated from data; posteriors are derived from neighbors following the finetuning steps. The predictive distribution with posterior is used for the final predictions.

## 2.2 Joint Meta Learning

Above, we assume access to given $M$ and $f_\phi$, in this section, we describe our meta-learning mechanism to train them jointly. The training strategy of Meta-Neighborhoods resembles MAML [6] in that both adopt a nested optimization procedure, which involves an inner loop update and an outer loop update in each iteration. Note that in contrast to MAML, we are solving a general discriminative task rather than a few-shot task. Given a batch of training data $\{x_i, y_i\}_{i=1}^B$ with a batch size $B$, in the inner loop we finetune the initial parameter $\phi$ to $\phi_i$ in a similar fashion to (1). With $\phi$ individually finetuned for each training data $x_i$ using its corresponding neighborhoods, we then jointly train the model parameter $\phi$, the dictionary $M$ as well as the inner loop learning rate $\alpha$ in the outer loop using the following meta-objective function

$$\mathcal{L}^{\text{meta}}(\phi, M, \alpha) = \frac{1}{B} \sum_{i=1}^B L(f_{\phi_i}(x_i), y_i) = \frac{1}{B} \sum_{i=1}^B L(f_{\phi - \alpha \nabla_\phi \mathcal{L}_i^{\text{inner}}}(x_i), y_i), \tag{2}$$

where $\mathcal{L}_i^{\text{inner}}(\phi) = \sum_{j=1}^S \omega(x_i, k_j) L(f_\phi(k_j), v_j)$ according to (1). We set $\alpha$ to be a learnable scalar or diagonal matrix. $\mathcal{L}^{\text{meta}}$ encourages learning shared $\phi$, $M$, and $\alpha$ that are widely applicable for data with the same distribution as the training data. An overview of our model is shown in Fig. 2.

Parameter $\phi$ serves as initial weights that can be quickly adapted to a specified neighborhood. This meta training scheme effectively tackles the overfitting problem caused by the limited number of finetuning instances, as it explicitly optimizes the generalization performance after finetuning.

For high-dimensional inputs such as images, learning $k_j$ in the input space could be prohibitive. Therefore, we employ a feature extractor $\mu_\theta$ to extract the feature embedding $z_i = \mu_\theta(x_i)$ for each

$x_i$ and learn $k_j$ in the embedding space. We accordingly modify (1) to

$$\phi_i \leftarrow \phi - \alpha \nabla_\phi \sum_{j=1}^{S} \omega(\mu_\theta(x_i), k_j) L(f_\phi(k_j), v_j), \tag{3}$$

where the attention function $\omega$ is employed in embedding space. The meta-objective is accordingly modified as $\mathcal{L}^{\text{meta}}(\phi, \theta, M, \alpha) = \frac{1}{B} \sum_{i=1}^{B} L(f_{\phi_i}(\mu_\theta(x_i)), y_i)$. We train $\theta$ and other learnable parameters jointly. Note that the model without a feature extractor can be viewed as a special case where $\mu_\theta$ is an identity mapping. The pseudocode of our training algorithm is given in Algorithm 1.

It is also desirable to adjust $\mu_\theta$ per-instance. However, when $\mu_\theta$ is a deep convolution neural network, tuning the entire feature extractor $\mu_\theta$ is computationally expensive. Inspired by FiLM [28], we propose instance-wise FiLM (iFiLM) that adjusts the batch normalization layers individually for each instance. Suppose $a^l \in \mathbb{R}^{B \times C^l \times W^l \times H^l}$ is the output of the $l_{th}$ Batch Normalization layer $\mathbf{BN}^l$, where $B$ is batch size, $C^l$ is the number of channels, $W^l$ and $H^l$ are the feature map width and height. Along with each $\mathbf{BN}^l$, we define a learnable dictionary $M^l = \{k_j^l, \gamma_j^l, \beta_j^l\}_{j=1}^{S_l}$ of size $S^l$. $k_j^l$ are the keys used for querying. $\gamma_j^l, \beta_j^l \in \mathbb{R}^{C^l}$ represent the scale and shift parameters used for adaptation respectively. When querying $M^l$, the outputs $a^l$ are first aggregated across their spatial dimensions using global pooling, i.e. $g^l = \mathbf{GlobalAvgPool}(a^l) \in \mathbb{R}^{B \times C^l}$. Then, the instance-wise adaptation parameters $\widehat{\gamma_i^l}$ and $\widehat{\beta_i^l}$ are computed as

$$\widehat{\gamma_i^l} = \sum_{j=1}^{S^l} \omega(g_i^l, k_j^l)\gamma_j^l \in \mathbb{R}^{C^l} \qquad \widehat{\beta_i^l} = \sum_{j=1}^{S^l} \omega(g_i^l, k_j^l)\beta_j^l \in \mathbb{R}^{C^l}, \tag{4}$$

where $\omega$ is defined as in (1) and $i \in \{1, 2, \ldots, B\}$. Following FiLM [28], each individual activation $a_i^l$ is then adapted with an affine transformation $\widehat{\gamma_i^l} \cdot a_i^l + \widehat{\beta_i^l}$. Note the transformation is agnostic to spatial position, which helps to reduce the number of learnable parameters in the dictionary $M$.

## 2.3 Other Details and Considerations

In this section, we discuss further implementation details. We also motivate our method from the perspective of $k$-nearest neighbor (KNN).

**Similarity Metrics**   To implement the attention function $\omega$ in (1)(3)(4), we need a similarity metric to compare a input $x_i$ with each key $k_j$. We try two types of metrics, cosine similarity and negative Euclidean distance. The similarities of $x_i$ to all keys are normalized using a softmax function with a temperature parameter $T$ [14], i.e.,

$$\omega(x_i, k_j) = \frac{\exp(\text{sim}(x_i, k_j)/T)}{\sum_{s=1}^{S} \exp(\text{sim}(x_i, k_s)/T)}, \tag{5}$$

where $\text{sim}(\cdot)$ represents the similarity metric.

**Initialization of the Dictionary**   Since we use similarity-based attention function $\omega$ in (3), we would like to initialize the key $k_j$ to have a similar distribution to $z_i = \mu_\theta(x_i)$, otherwise, $k_j$ cannot receive useful training signal at early training steps. To simplify the initialization, we follow [8] to remove the non-linear function (e.g. ReLU) at the end of $\mu_\theta$ so that features extracted by $\mu_\theta$ are approximately Gaussian distributed. With this modification, we can simply initialize $k_j$ with Gaussian distribution.

**Cosine-similarity Based Classification**   Since the model $f_\phi$ is finetuned using the learned dictionary in the inner loop, the quality of the dictionary has a significant impact on final performance. Typical neural network classifiers employ dot product to compute the logits, where the magnitude and direction could both affect the results. Therefore, the model needs to learn both the magnitude and the direction of $k_j$. To alleviate the training difficulty, we eliminate the influence of magnitude by using a cosine similarity classifier, where only the direction of $k_j$ can affect the computation of logits. Cosine similarity classifiers have been adopted to replace dot product classifiers for few-shot learning [8, 2] and robust classification [40].

**Relationship to KNN** Below, we show that Meta-Neighborhoods can be derived as a direct generalization of KNN under a multi-task learning framework. Considering a regression task where the regression target is a scalar, the standard view of KNN is as follows. First, aggregate the $k$-nearest neighbors of a query $\tilde{x}_i$ from the training set $\mathcal{D}$ as $\mathbf{N}(\tilde{x}_i) = \{(x_j, y_j)\}_{j=1}^k \subset \mathcal{D}$. Then, predict an average of the responses in the neighborhood: $\hat{y} = \frac{1}{k} \sum_{j=1}^k y_j$.

Instead of simply performing an average of responses in a neighborhood, we frame KNN as a solution to a multi-task learning problem with tasks corresponding to individual neighborhoods as follows. Here, we take each query (test) data $\tilde{x}_i$ as a single task, $\mathcal{T}_i$. To find the optimal estimator on the neighborhood $\mathbf{N}(\tilde{x}_i) = \{(x_j, y_j)\}_{j=1}^k$, we optimize the following loss $\mathcal{L}_{\mathcal{T}_i}(f_i) = \frac{1}{k} \sum_{j=1}^k L(f_i(x_j), y_j)$ where $L$ is a supervised loss, and $f_i$ is the estimator to be optimized. For example, for MSE-based regression the loss for each task is $\mathcal{L}_{\mathcal{T}_i}(f_i) = \frac{1}{k} \sum_{j=1}^k (f_i(x_j) - y_j)^2$. If one takes $f_i$ to be a constant function $f_i(\eta_j) = C_i$, then the loss is simply $\mathcal{L}_{\mathcal{T}_i}(f_i) = \frac{1}{k} \sum_{j=1}^k (C_i - \zeta_j)^2$, which leads to an optimal $f_i(\tilde{x}_i) = C_i = \frac{1}{k} \sum_{j=1}^k \zeta_j$, the same solution as traditional KNN. Similar observations hold for classification. *Thus, given neighborhood assignments, one can view KNN as solving for individual tasks in the special case of a constant estimator $f_i(x_j) = C_i$.*

With the multi-task formulation of KNN, we can generalize KNN to derive our Meta-Neighborhoods method by considering a non-constant estimator as $f_i$. For instance, one may take $f_i$ as a parametric output function $f_{\phi_i}$ (e.g. a linear model or neural networks), and finetune the parameter $\phi$ to $\phi_i$ for a data $x_i$ according to the loss on neighborhood $\mathbf{N}(x_i)$. Instead of fitting a single label on the neighborhood, a parametric approach attempts to fit a richer (e.g. linear) dependency between input features and labels for the neighborhood. In addition, the multi-task formulation gives rise to a way of constructing meta-learning episodes. Also, we learn both neighborhoods and the function $f_\phi$ jointly in our Meta-Neighborhoods framework.

## 3 Related Work

**Memory-augmented Neural Networks** Augmenting neural network with memory has been studied in the sentinel work Neural Turing Machine [10], where a neural network can read and write an external memory to record and change its state. Recent works that utilize the memory modules generally fall into two categories. One category modifies the memory modules according to hand-crafted rules. For instance, previous works tackling few-shot classifications add a new slot to the memory when the label of a given data does not match the labels of its $k$-nearest neighbors from the memory [1] or the given data is misclassified [29]. [35] adopts a fixed-size memory that acts as a circular buffer for life-long learning. Another category uses a fully-differentiable memory module and trains it together with neural networks by gradient descent. This type of memory has been explored for knowledge-based reasoning [11], sequential prediction [36] and few-shot learning [17, 32]. Our work also utilizes a differentiable memory but is used to capture local manifold and improve the general discriminative learning performance.

**Meta-Learning** Representative meta-learning algorithms can be roughly categorized into two classes: initialization based and metric-learning based. Initialization based methods, such as MAML [6], learn a good initialization for model parameters so that several gradient steps using a limited number of labeled examples can adapt the model to make predictions for new tasks. To further improve flexibility, Meta-SGD [23] learns coordinate-wise inner learning rates, and curvature information is considered in [27] to transform the gradients in the inner optimization. Metric-learning based methods focus on using a distance metric on the feature space to compare query set samples with labeled support set samples. Examples include cosine similarity [39] or Euclidean distance [33] to support examples. A learned relation module is employed in [37].

Our model is in a similar vein to the initialization based method: each test sample can be regarded as a new task, and we meta-learn a dictionary that adapts the initial model to a local neighborhood by finetuning over queried neighbors. A recent work Meta AuXiliary Learning (MAXL) also explores meta-learning techniques to improve classification performance, where a label generator is meta learned to generate auxiliary labels so that the auxiliary task trained together with the primary classification task can improve the primary performance.

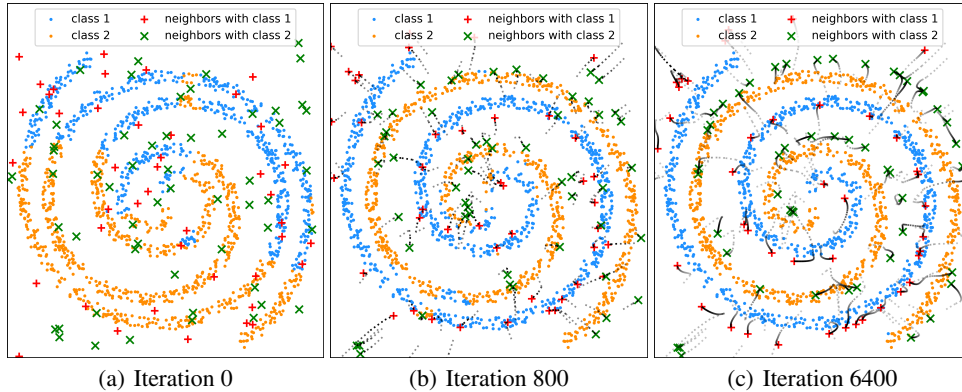

| (a) Iteration 0 | (b) Iteration 800 | (c) Iteration 6400 |

Figure 3: Evolution of learnable neighbors and classification results on the test data during training. Two classes are two spirals. Binary predictions for the test set are shown as blue and yellow points. Learnable neighbors are first randomly initialized in (a), then optimized in (b) (c). The dotted black lines are the trajectories of learnable neighbors through training iterations. A video of the optimization process is given in the supplemental materials.

# 4 Experiments

In this section, we conduct experiments for both classification and regression tasks. To demonstrate the benefits of making predictions in local neighborhoods, we compare to the *vanilla* model where the same network architecture is used but without the learnable neighborhoods. We also compare to MAXL [24] for classification task.

## 4.1 Toy Example: Binary Classification of the Concentric Spiral Dataset

To investigate the behavior of the induced neighbors and how they assist the parametric model to make predictions, we first classify on a 2D toy dataset. In this binary classification task, points from two classes are placed in concentric spirals with a non-linear decision boundary. Although a linear classifier is incapable of capturing this decision boundary, we show that tuning a linear classifier with induced neighbors gives an overall non-linear classifier. In addition, our learned neighborhoods capture the critical manifold structure and concentrate at boundary cases; we also observe semantically relevant learned neighbors in higher dimensions (see Appendix A.8 and A.9)

In Fig. 3, we visualize the evolution of the learned neighbors and the decision boundary. The learnable neighbors (shown in green and red markers for two classes respectively) are first initialized with random keys and values. As we train the model and the neighbors, the learned neighbors are gradually driven to important manifold locations as in Fig. 3 (b) and (c). After training, the linear classifier adapted to local neighborhoods can accurately classify test examples. We use 100 induced neighbors. The labels of neighbors (values in the dictionary) are fixed after random initialization for illustration purposes. The 2D locations of neighbors (keys in the dictionary) are updated with the model. We use negative Euclidean distance as the similarity metric in (5) and set $T$ to 0.1.

## 4.2 Image Classification

In this section, we evaluate 9 datasets with different complexities and sizes: MNIST [21], MNIST-M[7], PACS[22], SVHN [9], CIFAR-10 [19], CIFAR-100 [19], CINIC-10 [3], Tiny-ImageNet [31] and ImageNet [4]. Dataset details and preprocessing methods are given in Appendix A.1.

Our models are compared to two baselines: *vanilla*, a traditional parametric ConvNet with the same architecture as ours but without the learnable dictionary, and MAXL [24], where an auxiliary label generator is meta-learned to enhance the primary classification tasks. For MNIST, MNIST-M, SVHN and PACS, a 4-layer ConvNet is selected as the feature extractor $\mu_\theta$. For the other four datasets, three deep convolutional architectures, DenseNet40-BC [15], ResNet29, and ResNet56 [13], are used as

Table 1: The classification accuracies of our model and the baselines. "MN" denotes Meta-Neighborhoods. Results from three individual runs are reported and the best performance is marked as bold. Given that backbones like ResNet-56 are strong, our consistent improvement is notable.

| Datasets | *vanilla* | MAXL | ours | | |
|---|---|---|---|---|---|
| | | | *vanilla*+iFiLM | MN | MN+iFiLM |
| **Backbone:** 4-layer ConvNet | | | | | |
| MNIST | 99.44±0.03% | 99.60±0.02% | 99.40±0.02% | **99.62±0.03%** | 99.58±0.03% |
| SVHN | 93.02±0.12% | 94.06±0.10% | 93.95±0.12% | 94.46±0.09% | **94.92±0.09%** |
| MNIST-M | 96.18±0.05% | 96.85±0.06% | 96.99±0.07% | 96.55±0.04% | **97.40±0.05%** |
| PACS | 92.55±0.08% | 94.85±0.12% | 94.45±0.09% | 95.19±0.10% | **95.22±0.09%** |
| **Backbone:** DenseNet40-BC | | | | | |
| CIFAR-10 | 94.53±0.10% | 94.83±0.09% | 94.87±0.08% | 95.04±0.11% | **95.22±0.09%** |
| CIFAR-100 | 73.92±0.12% | 75.64±0.14% | 74.66±0.13% | 76.32±0.16% | **76.96±0.14%** |
| CINIC-10 | 84.92±0.07% | 85.42±0.07% | 85.11±0.08% | 85.73±0.10% | **86.02±0.07%** |
| Tiny-ImageNet | 49.28±0.18% | 50.94±0.16% | 50.86±0.14% | 53.27±0.18% | **54.36±0.15%** |
| **Backbone:** ResNet-29 | | | | | |
| CIFAR-10 | 95.06±0.10% | 95.31±0.09% | 95.17±0.10% | 95.56±0.09% | **95.58±0.10%** |
| CIFAR-100 | 76.51±0.15% | 77.94±0.12% | 77.16±0.14% | 78.84±0.14% | **79.84±0.11%** |
| CINIC-10 | 86.03±0.08% | 86.34±0.06% | 86.64±0.06% | 86.86±0.08% | **87.35±0.09%** |
| Tiny-ImagNnet | 54.82±0.17% | 56.29±0.14% | 55.59±0.17% | 57.36±0.15% | **57.94±0.14%** |
| **Backbone:** ResNet-56 | | | | | |
| CIFAR-10 | 95.73±0.08% | 96.06±0.07% | 96.08±0.08% | 96.36±0.07% | **96.40±0.06%** |
| CIFAR-100 | 79.64±0.13% | 80.36±0.13% | 80.04±0.12% | 80.58±0.10% | **80.90±0.12%** |
| CINIC-10 | 88.21±0.07% | 88.30±0.05% | 88.57±0.07% | 88.61±0.06% | **88.99±0.07%** |
| Tiny-ImageNet | 57.92±0.12% | 58.94±0.16% | 58.31±0.15% | 60.05±0.12% | **60.78±0.13%** |
| ImageNet | 48.41±0.14% | 48.83±0.16% | 52.03±0.12% | 51.85±0.12% | **54.23±0.13%** |

$\mu_\theta$. $f_\phi$ is implemented as a cos-similarity based classifier with one linear layer for both *vanilla* and our models. Experiment details for our models and baselines are provided in Appendix A.2.

**Results** Table 1 compares the test accuracy to baselines. We show that both Meta-Neighborhoods (MN) and iFiLM (*vanilla*+iFiLM) can improve over *vanilla*. The best performance is achieved when combining MN and iFiLM (MN+iFiLM), which outperforms the *vanilla* and MAXL baselines across several network architectures and different datasets. This indicates Meta-Neighborhoods and iFiLM are complementary and it is beneficial to adjust both $f_\phi$ and $\mu_\theta$ per-instance. Our method is also effective at PACS and MNIST-M datasets that contain significant domain shifts.

Note that backbones like ResNet-56 are *already powerful* for these datasets and there is limited room for improvement over the *vanilla* model. For instance, employing ResNet-110 instead of ResNet-56 *only gives 0.14% and 0.40% further improvements* on CIFAR-10 and CIFAR-100, but at the expense of doubling the number of parameters. Yet Meta-Neighborhoods still consistently achieve greater improvements over *vanilla* than previous SOTA meta-learning method MAXL [24]. Compared to *vanilla* models, Meta-Neighborhoods with the same backbone architecture contains extra trainable parameters stored in the dictionary $M$. However, as discussed in Appendix A.3, the performance boost in our paper originates from adjusting models using neighbors, rather than a naive increase in the number of parameters.

Since we implement $f_\phi$ as a cosine similarity classification layer, $\phi$ can be regarded as the prototypes for each class. To verify that finetuning over neighborhoods helps with the classification, we compare the cosine similarity between the extracted feature $z_i$ and its corresponding ground-truth prototypes $\phi[y_i]$ before and after finetuning, where $y_i$ is the class label for $z_i$. From Fig. 4, we can see that the cosine similarities increase after finetuning for most test examples, which indicates better predictions after finetuning.

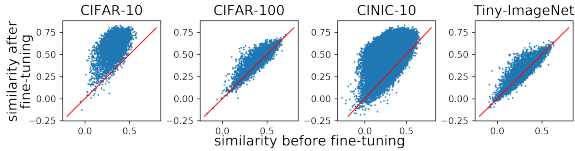

Figure 4: Cosine similarities between features and their corresponding ground-truth class prototypes. Each blue point denotes a testing sample. We expect most samples locate above the red lines, meaning larger similarities after finetuning.

We conduct ablation studies for $S$ and $T$ in Appendix A.4. Ablations for the number of inner loop finetuning steps and different forms of $\alpha$ (scalar or diagonal) are provided in Appendix A.5.

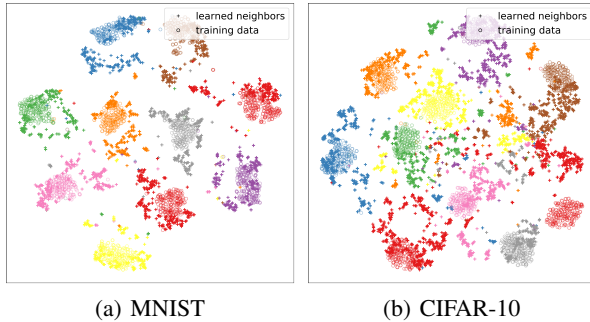

(a) MNIST       (b) CIFAR-10

Figure 5: t-SNE visualization of the learned neighbors and training data on MNIST and CIFAR-10. Learned neighbors are marked as "+" and real training data are marked as "o". The class information is represented by colors. Please zoom in to see the differences between "+" and "o".

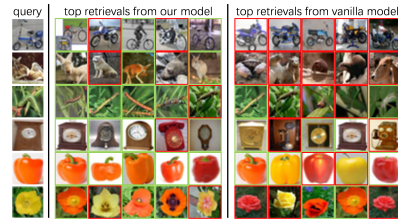

Figure 6: Sub-category image retrieval quality of our model and the *vanilla* model. Correct retrievals have green outlines and wrong retrievals have red outlines.

Additional experiments for *vanilla* models trained with dot-product output layer and SGD are provided in Appendix A.6. We also discuss the inference speed of our model in Appendix A.7.

**Analysis of Learned Neighbors** In Fig. 5, we use t-SNE [26] to visualize the 2D embeddings of the learned neighbors (marked as "+") along with the training data (marked as "o") on CIFAR-10 and MNIST. The 2D embeddings of training data are computed on the features $z_i = \mu_\theta(x_i)$, and embeddings of learned neighbors are computed on keys $k_j$. Classes of the learned neighbors are inferred from values $v_j$. It is shown that our model learns neighbors beyond the training set as the learned neighbors do not completely overlap with the training data, and the learned neighbors represent "hard cases" around class boundaries to assist our model making better predictions. It is also interesting to note that this follows the same trend as our toy example in Fig. 3.

We further investigate whether the learned neighbors are semantically meaningful by retrieving their 5-nearest neighbors from the test set. As shown in Appendix A.8, the retrieved 5-nearest neighbors for each learned neighbor not only come from the same class, but also represent a specific sub-category concept. In Appendix A.9, we quantitatively show that our method has superior sub-category discovery performance than *vanilla* on CIFAR-100: our method achieves 63.3% accuracy on the 100 fine-grained classes while *vanilla* only achieves 59.28%. This indicates our learned neighbors can preserve fine-grained details that are not explicitly given in the supervision signal. Qualitative results are shown in Fig. 6.

### 4.3 Regression

We use five publicly available datasets with various sizes from UCI Machine Learning Repository [5]. For regression tasks, we found learning neighbors in the input space yields better performance

Table 2: Test MSE of our model, kNN and the *vanilla* baseline on five datasets. $n$ and $d$ respectively denote the dataset size and the data dimension.

| Datasets | $n$ | $d$ | kNN | *vanilla* | Meta-Neighborhoods |
|---|---|---|---|---|---|
| music | 515345 | 90 | 0.6812±0.0062 | 0.6236±0.0056 | **0.6088±0.0050** |
| toms | 28179 | 96 | 0.0602±0.0083 | 0.0594±0.0080 | **0.0531±0.0073** |
| cte | 53500 | 384 | 0.00134±0.00023 | 0.00121±0.00022 | **0.00109±0.00015** |
| super | 21263 | 80 | 0.1126±0.0061 | 0.1132±0.0060 | **0.1077±0.0068** |
| gom | 1059 | 116 | 0.5982±0.0521 | 0.5949±0.0515 | **0.5681±0.0563** |

compared to learning neighbors in the feature space. As a result, the feature extractor $\mu_\theta$ is implemented as an identity mapping function. We compare our model to kNN and *vanilla* baseline using the mean square error (MSE). The *vanilla* baseline is a multilayer perceptron for regression. We searched for the best network configuration for the *vanilla* model on every dataset by varying the number of layers in $\{2, 3, 4, 5\}$ and the number of neurons at each layer in $\{32, 64, 128, 256\}$. For each dataset, our model uses the same network architecture to *vanilla*. Model details, training details, and hyperparameter settings are given in Appendix B. 5-fold cross-validation is used to report the results in Table 2. Our model has lower MSE scores compared to the *vanilla* model across the five datasets. The results of Meta-Neighborhoods and *vanilla* are statistically different based on paired Student's t-test with a significance of 0.05. We found that naively increasing the model complexity for *vanilla* baseline can not further improve its performance due to over-fitting, but our method can as it takes advantage of non-parametric neighbor information.

## 5 Conclusion

In this work, we introduced Meta-Neighborhoods, a novel meta-learning framework that adjusts predictions based on learnable neighbors. It is interesting to note that in addition to directly generalizing KNN, Meta-Neighborhoods provides a learning paradigm that aligns more closely with human learning. Human learning jointly leverages previous examples both to shape the perceptual features we focus on and to pull relevant memories when faced with novel scenarios [20]. In much the same way, Meta-Neighborhoods use feature-based models that are then adjusted by pulling memories from previous data. We show through extensive empirical studies that Meta-Neighborhoods improve the performance of already strong backbone networks like DenseNet and ResNet on several benchmark datasets. In addition to providing a greater gain in performance than previous state-of-the-art meta-learning methods like MAXL, Meta-Neighborhoods also works both for regression and classification, and provides further interpretability.

## Broader Impact

Any general discriminative machine learning model runs the risk of making biased and offensive predictions reflective of training data. Our work is no exception as it aims at improving discriminative learning performance. To reduce these negative influences to the minimum possible extent, we only use standard benchmarks in this work, such as CIFAR-10, Tiny-ImageNet, MNIST, and datasets from the UCI machine learning repository.

Our work does impose some privacy concerns as we are learning a per-instance adjusted model in this work. Potential applications of the proposed model include precision medicine, personalized recommendation systems, and personalized driver assistance systems. To keep user data safe, it is desirable to only deploy our model locally.

The induced neighbors in our work, which are semantically meaningful, can also be regarded as fake synthetic data. Like DeepFakes, they may also raise a set of challenging policy, technology, and legal issues. Legislation regarding synthetic data should take effect and the research community needs to develop effective methods to detect these synthetic data.

## Acknowledgments and Disclosure of Funding

This work was supported in part by NIH 1R01AA02687901A1.

## Footnotes

[1]The code is available at https://github.com/lupalab/Meta-Neighborhoods

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
