[Supplementary Material]

# A Experiment Details for Classification

## A.1 Dataset Details

All datasets except ImageNet were resized to resolution 32×32 to facilitate fast experimentation. ImageNet was resized to resolution 64×64. CIFAR-10/100 are image classification datasets containing a training set of 50K and a testing set of 10K 32×32 color images across the 10/100 classes. CINIC-10 has 270K 32×32 images across 10 classes equally split into three parts for training, validation, and testing. Tiny-ImageNet has a training set of 100K and a testing set of 10K 64×64 images across the 200 classes. PACS contains domain shifts (Art painting, Cartoon, Photo and Sketch), and we train on all domains to test our model's ability to fit on all domains. PACS is split as 9K/1K/10K for training/validation/testing. MNIST-M also contains domain shifts due to the random background. MNIST-M is split as 55K/5K/10K for training/validation/testing.

## A.2 Implementation Details

We find adaptive learning rate optimizers like Adam [18] are more effective than SGD at optimizing the dictionary, as these optimizers can compute individual adaptive learning rates for different parameters. In all the experiments, we adopt AdamW [25], a variant of Adam [18] with correct weight decay mechanism and better generalization capability.

Our models are trained by AdamW with weight decay rate 7.5e-5, an initial learning rate of 1e-3 and batch size 128. For CIFAR-10 and CIFAR-100, we train for 400 epochs with learning rate reduced to 1e-4 at epoch 300. For CINIC-10 and Tiny-ImageNet, models are trained for 350 epochs with the learning rate reduced to 1e-4 at epoch 250. We train *vanilla* baselines by AdamW in the same way as our models.

We initialize $k_j$ and $v_j$ with Gaussian distribution $\mathcal{N}(0, 0.01)$, and apply a softmax over $v_j$ to make it a probability distribution over multiple classes. Cosine similarity is adopted as the similarity metric in (5). We set $T$ in (5) to 0.2. For CIFAR-10 and CINIC-10, the number of dictionary entries $S$ is set to 5000, and for CIFAR-100 and Tiny-Imagenet $S$ is set to 10000 in the main results. For the results in Table 1, we set the number of inner loop steps to 1 and set $\alpha$ to be a learnable scalar. Ablation studies of the number of inner loop finetuning steps and different forms of $\alpha$ (scalar or diagonal) are given in Appendix A.5.

When optimizing the dictionary at the early stage, some entries might always receive larger gradients than others due to the random initialization, which causes the model exploiting only part of the dictionary entries. To boost the diversity of used entries, we randomly dropout 50% of the entries during training. This also helps to prevent overfitting and improve training speed.

For models (Densenet40-BC, ResNet) with iFiLM, we learn a learnable fix-sized dictionary $M_l = \{k_j^l, \gamma_j^l, \beta_j^l\}_{j=1}^{S_l}$ with every batch normalization layer, where $S_l$ is set to 10 for all $M_l$.

We ran MAXL using code provided in [24] and performed extensive hyperparameter tuning to report the best results. We run MAXL with SGD of a learning rate of 0.1 with momentum 0.9 and weight decay $5 \times 10^4$, and hierarchies to set to 5 as recommended in MAXL. We also try a larger learning rate of 0.01. We dropped the learning rate by half for every 50 epochs with a total of 200 epochs.

|     (a) ResNet-29     |     (b) ResNet-56     |     (c) DenseNet40-BC     |

Figure 7: Validation accuracies of *vanilla* models with (shown in orange) and without (shown in blue) extra parameters on CIFAR-100 across three backbone architectures.

### A.3  *vanilla* Models with Extra Parameters

Compared to *vanilla* models, our best model (MN+iFiLM) with the same backbone architectures contain $S \times (d + C)$ extra trainable parameters in the dictionary $M$ and $3 \times S^l \times C^l$ extra trainable parameters in $M^l$, where $S$ is the number of dictionary entries in $M$, $d$ is the dimension of the feature vector, $C$ is the number of classes, $S^l$ is the number of dictionary entries in $M^l$ and $C^l$ is the number of feature map channels. Since we set a small $S^l = 10$, extra parameters in $M^l$ is negligible compared to the remaining model parameters. Therefore we only consider the extra parameters in $M$ in this section.

To investigate whether *vanilla* models can also benefit from more parameters as our model, we add an extra fully connected layer with the same number of extra parameters to *vanilla* models. According to Fig. 7, *vanilla* models with extra parameters have inferior validation accuracy than *vanilla* models without extra parameters across three backbone architectures on CIFAR-100. The same observation holds for other datasets. Therefore, *vanilla* models can not benefit from more parameters as our model, and the performance boost in our paper originates from finetuning using neighbors, rather than a naive increase in parameters.

We show in Table 1 that our method consistently improves performance regardless of the choice of backbones, indicating our method can further improve over an even bigger model.

Note that our method is particularly helpful for low-capacity models, which usually handle simpler tasks like MNIST/SVHN classification and regression. In the regression experiment in Section 4.3, we show that naively increasing the model capacity can't effectively further improve performance on these simple tasks due to over-fitting, but our method can as it takes advantage of non-parametric neighbor information.

### A.4  Ablation Study of $S$ and $T$

We investigate the impact of hyperparameters, $S$ and $T$ on CIFAR-100 using ResNet-29. As shown in Fig. 8, the testing accuracy increases with the increase of the number of dictionary entries $S$ (with $T$ set to 0.2), which indicates a better finetuning of $\phi$ with the help of more learnable neighbors. The temperature $T$ controls the "peakiness" of the similarity distribution in (5). It is set to a fixed value rather than learned in all experiments. If we enable $T$ to be learnable, it always grows to a small value, which makes the model only pay attention to a small number of entries and leads to over-fitting. On the other hand, if $T$ is too large, the model will pay uniform attention to all entries in $M$, which leads to under-fitting. According to Fig. 8, the optimal range of $\frac{1}{T}$ is [3,7] (with $S$ set to 10000).

### A.5  Ablation Study of the Number of Inner Loop Steps and the Form of $\alpha$

As shown in Table 3, we found that implementing the inner loop learning rate $\alpha$ as a learnable scalar usually gives better performance. The majority of our results are better when setting the number inner-loop finetuning steps to 1. As we learn $\alpha$ rather than set it to a fixed value, our model can achieve good performance in only one step of finetuning.

We also investigate how Meta-Neighborhoods performs without finetuning at test time. With DenseNet40, our pre-tuned and post-tuned models achieve 90%/95% on cifar10 and 69%/76% on cifar100, indicating it is beneficial to finetune the model. This is also substantiated by Fig. 4. This

Figure 8: Ablation studies of $S$ and $T$ on CIFAR-100.

Table 3: The classification accuracies of our model and the baselines. "ft" in our methods denotes how many finetuning steps are used in the inner loop. "S" in our methods denotes using a scalar inner loop learning rate, while "D" denotes using a diagonal matrix inner loop learning rate.

| Datasets | vanilla | | | | MAXL | Meta-Neighborhoods (ours) | | | |
|---|---|---|---|---|---|---|---|---|---|
| | dot-sgd | dot-adamw | cos-sgd | cos-adamw | | ft:1+S | ft:1+D | ft:3+S | ft:3+D |
| **Backbone:** 4-layer ConvNet | | | | | | | | | |
| MNIST | 99.39% | 99.47% | 99.42% | 99.44% | 99.60% | **99.62%** | 99.45% | 99.50% | 99.55% |
| SVHN | 93.01% | 93.12% | 92.93% | 93.02% | 94.06% | **94.46%** | 93.95% | 94.05% | 94.05% |
| **Backbone:** DenseNet40-BC | | | | | | | | | |
| CIFAR-10 | 94.56% | 94.46% | 94.52% | 94.53% | 94.83% | 95.04% | 94.79% | 95.08% | **95.12%** |
| CIFAR-100 | 73.85% | 73.68% | 74.08% | 73.92% | 75.64% | 76.32% | **77.20%** | 76.04% | 76.42% |
| CINIC-10 | 85.13% | 85.02% | 85.10% | 84.92% | 85.42% | 85.73% | **85.76%** | 85.51% | 85.21% |
| Tiny-Imagenet(32×32) | 49.32% | 49.21% | 49.40% | 49.28% | 50.94% | **53.27%** | 53.16% | 52.88% | 52.61% |
| **Backbone:** ResNet-29 | | | | | | | | | |
| CIFAR-10 | 94.91% | 94.96% | 95.02% | 95.06% | 95.31% | **95.56%** | 95.28% | 95.36% | 95.26% |
| CIFAR-100 | 76.65% | 76.72% | 76.70% | 76.51% | 77.94% | **78.84%** | 78.20% | 78.04% | 78.40% |
| CINIC-10 | 85.86% | 85.91% | 85.96% | 86.03% | 86.34% | **86.86%** | 86.41% | 86.38% | 86.51% |
| Tiny-Imagenet(32×32) | 54.79% | 54.67% | 54.97% | 54.82% | 56.29% | 57.36% | 56.93% | **57.64%** | 57.27% |
| **Backbone:** ResNet-56 | | | | | | | | | |
| CIFAR-10 | 95.64% | 95.83% | 95.71% | 95.73% | 96.06% | **96.36%** | 96.32% | 96.28% | 96.04% |
| CIFAR-100 | 79.54% | 79.68% | 79.78% | 79.64% | 80.36% | 80.58% | **80.66%** | 80.20% | 80.14% |
| CINIC-10 | 88.03% | 88.15% | 87.90% | 88.21% | 88.30% | **88.61%** | 88.47% | 88.42% | 88.38% |
| Tiny-Imagenet(32×32) | 57.79% | 57.95% | 57.89% | 57.92% | 58.94% | **60.05%** | 59.20% | 59.85% | 59.88% |

bad result without finetuning at the testing stage is unsurprising because the model is used differently at the training stage and the testing stage.

## A.6 Ablation Study of *vanilla* Models

To ensure that the modifications (using cos-similarity output layer and AdamW optimizer) that we made to the commonly-used *vanilla* model trained by SGD with dot-product output layer do not deteriorate the performance, we also report the accuracies of *vanilla* models with either dot-product or cos-similarity output layer and trained either by AdamW or SGD in Table 3. All these four *vanilla* baselines have similar performance that is worse compared to our method.

## A.7 Inference Speed

Due to the neighbor searching process and finetuning process, our method is slower at the testing time compared to the *vanilla* testing process which only requires a single feed-forward propagation. However, our method is only approximately 2 times slower than the vanilla models due to the following reasons: (1) our searching space is small (only 5000 neighboring points) (2) the attention calculation and the finetuning step are parallelized efficiently across multiple GPU threads (3) for the classification task, we only finetune the parameters of the classification layer rather than the whole model (4) we only finetune for a small number of steps (1 or 3).

## A.8 Visualizing Learned Neighbors by Retrieving Real Neighbors

We further investigate whether the learned neighbors are semantically meaningful by retrieving their 5-nearest neighbors from the test set.

Examples on CIFAR-10 and MNIST are respectively shown in Fig. 9 and 10. We found most entries can retrieve consistent neighbors. It is shown that the retrieved 5-nearest neighbors for each learned neighbor not only come from the same class, but also represent a specific sub-category concept. For instance, both of the entries on the fifth row of Fig. 9 represent "ship", but the first represents "steamship" while the second represents "speedboat".

## A.9 Sub-category Discovery

To quantitatively measure the sub-category discovery performance, we train our model ($S$ is set to 5000) and the *vanilla* cos-adamw model with the same Densenet40 backbone on CIFAR-100 only using its coarse-grained annotations (20 classes), and evaluate the classification accuracy on the fine-grained 100 categories using KNN classifiers, which is called induction accuracy as in [16]. For the *vanilla* model, the KNN classifier uses the feature $z_i = \mu_\theta(x_i)$, while for our model, the classifier

Figure 9: 5-nearest neighbors of 12 dictionary entries retrieved using $k_j$ in (3) from the CIFAR-10 test set. Entry indexes and entry classes inferred from $v_j$ are shown on the left of each group of images. By comparing the two entries on the same row, we discover that different entries represent different fine-grained sub-category concepts.

Figure 10: 5-nearest neighbors of 30 dictionary entries retrieved using $k_j$ in (3) from the MNIST test set. By comparing the three entries on the same row, we discover that different entries represent different fine-grained attributes such as stroke widths, character orientations and fonts.

uses the attention vector $\vec{\omega} = [\omega(z_i, k_1), \omega(z_i, k_2), ..., \omega(z_i, k_S)]$ over all entries in $M$. Our KNN and *vanilla*'s KNN achieve similar accuracy on the coarse 20 classes (80.18% versus 80.30%). However, on the fine-grained 100 classes, our KNN achieves 63.3% while the *vanilla*'s KNN only achieves 59.28%. This indicates our learned neighbors can preserve fine-grained details that are not explicitly given in the supervision signal. Examples of nearest neighbors retrieved are shown in Fig. 6.

## B   Experiment Details for Regression

We use five publicly available datasets with various sizes from UCI Machine Learning Repository: *music* (YearPredictionMSD), *toms*, *cte* (Relative location of CT slices on axial axis), *super* (Superconduct), and *gom* (Geographical Original of Music). All datasets are normalized dimension-wise to have zero means and unit variances.

For regression tasks, we found learning neighbors in the input space yields better performance compared to learning neighbors in the feature space. As a result, our model for regression only consists of an output network $f_\phi$ and a dictionary $M$. It is trained with the loss in (2). A learning rate of 1e-3 and a batch size of 128 are used, and the best weight decay rate is chosen for each dataset. The training stops if the validation loss does not reduce for 10 epochs. We initialize $k_j$ with Gaussian distribution $\mathcal{N}(0, 0.01)$ and $v_j$ with uniform distribution in the same range of the regression labels. Cosine similarity is adopted to implement the similarity metric in (5). We use 1000 dictionary entries and set $T$ to 0.1 based on the validation performance. Because there is no batch normalization layer in $f_\phi$, iFiLM is not used in this regression experiment.