[Reviews · NeurIPS 2020]

Review 1

Summary and Contributions: The paper proposes a method for adapting the parameters of a model at test time using a learned dictionary. It resembles MAML except that in the inner loop the loss due to each point in the support set (dictionary) is multiplied by a weight obtained by calculating the similarity between the query point and the corresponding point in the support set (dictionary). They also learn the learning rate like in Meta-SGD and do a FiLM style adaptation of the batch-normalization layer parameters.

Strengths: (i) The idea of combining single parameter initialization of a model with non-parametric approaches which use the local neighbourhood to adapt at test time is interesting. Though to me, a more natural extension would have been to think of approaches and experiments in the domain of transfer learning or few-shot learning where an example in the unseen domain is adapted based on how similar it is to examples from other domains seen previously. (ii) The paper is well written.

Weaknesses: Though the idea seems interesting, my main concern is the relatively weak experimental section: (i) Extremely minor improvements: Looking at the numbers in Table 1 and Table 2, if we include the confidence intervals MN is comparable to other techniques (though only the numbers based on mean are bolded in the table, which I find quite misleading). For example, in Table 1, 1st row MN numbers are comparable to MAXL but not bolded. In other cases, the improvements seem quite minor. Similarly, row 2, 3, 4, 5 in Table 2 are all in comparable range. (ii) Choice of datasets: I would have wanted to see results on some harder datasets like Imagenet and UCF-101 as in the MAXL paper. Also, UCI datasets for regression experiments seem pretty non-standard at these times. I would have loved to see results on few-shot benchmarks like Omniglot or mini-Imagnet too as they fit very nicely in the setting mentioned in the paper. (iii) Missing comparisons: Seeing the close resemblance of the approach with MAML and prototypical networks for the classification case, I think a comparison with these approaches is warranted. Also, I feel that the regression experiments should have a kNN baseline. (iv) Comparison with memory-networks: I think a comparison with previous works like [1] should be added. [1] Cai, Qi, Yingwei Pan, Ting Yao, Chenggang Yan, and Tao Mei. "Memory Matching Networks for One-Shot Image Recognition." arXiv preprint arXiv:1804.08281 (2018). *******************After Rebuttal************************* (i) I am happy that the authors performed experiments on Imagenet and their method seems better than vanilla model.However, I find many details missing: "Again, this improvement is larger than that achieved by MAXL. To facilitate experiments, we resize images to 64 × 64 resolution." As per my understanding, MAXL used a resolution of 32 X 32 and the numbers in their table compare a VGG-16 backbone. I don't see how the authors can make the statement that their improvment was better when compared to MAXL when clearly the numbers are not directly comparable at least from the results in Table 1 of MAXL. What is vanilla framework for their experiments? Why not show results of MAXL with the same configuration as the rest where there will be a fourth curve corresponding to MAXL? The lack of such details makes me a little skeptic. (ii) I do not agree that comparison with MAML is not warranted simply because their objective is not to do few shot learning. Their algorithm is heavily inspired from MAML and I think an ablation should be included. There is nothing in the way MAML is formulated which makes it applicable ONLY to few shot settings. It is applicable to all scenarios where any form of adaptation is involved. (iii) Based on values in Appendix A.3 and A.7, there is clearly an increase in number of parameters and the inference time is two times slower than vanilla models. There is no mention of inference speeds compared to MAXL. Why use a method which is slow and has more parameters especially when the improvements produced are minor ( please see above where I provide examples of cases where there is an overlap). However, after seeing results on Imagenet ( which I am still skeptic) and additional regression datasets and inclusion of kNN baselines, I am a bit more positive about the virtues of the method. I have increased my score to 5. I still do not feel entirely convinced to increase my score more than this.

Correctness: The formulation is correct. Claims: (i) (Abstract)- "To reduce memory and computation over- head, we propose induced neighborhoods that summarize the training data into a much smaller dictionary". I fail to see how memory and/or computation time is reduced. On the contrary, it seems there is an increase in the number of parameters as compared to other baselines. (ii) (Conclusion) - "In addition to providing a greater gain in performance than previous state-of-the-art meta-learning methods like MAXL, Meta-Neighborhoods also works both for regression and classification, and provides further interpretability" - as stated before, the experimental results show little or (none in regression) improvement in performance. I also didn't see how interpretability is improved as compared to MAXL.

Clarity: The paper is very well written and the figures are nice. I liked the video as well.

Relation to Prior Work: As stated earlier comparisons can be done with MAML, protonets and matching networoks.

Reproducibility: Yes

Additional Feedback: (i) Though very similar, I will be interested to see how the method performs when the formulation in (5) is replaced by scaled dot product attention as in equation 1 of [1]. (ii) I think an addition of few-shot learning benchmarks like mini-Imagenet and Omniglot and motivating with respect to transfer to unseen domains will make the paper stronger. [1] Vaswani, Ashish, Noam Shazeer, Niki Parmar, Jakob Uszkoreit, Llion Jones, Aidan N. Gomez, Łukasz Kaiser, and Illia Polosukhin. "Attention is all you need." In Advances in neural information processing systems, pp. 5998-6008. 2017.


Review 2

Summary and Contributions: This paper proposes a meta learning method for automatically generate psedo-NNs from data, and then use them to guide the discriminative learning for training data. To my knowledge, this is the first meta learning strategy for guiding the NN learning. The effectiveness of the proposed method is validated in both classification and regression tasks.

Strengths: The proposed meta-NN idea is rational and novel. The designed algorithm for generating selected NNs, learning rate, feature extractor variables is rational. Experiments for validating the effectiveness of the method are sufficient and convincing.

Weaknesses: Too many variables to be learned, making the method highly non-convex and hardly to be trained.

Correctness: The method should be rational, but some claims might not be that correct.

Clarity: Yes.

Relation to Prior Work: Clear.

Reproducibility: No

Additional Feedback: The proposed idea is interesting and novel to me, and should be rational. But my main concern is that there are too many variables required to be meta-trained, including the pseudo-NNS, learning rate, and feature extractor. I think this might impose too much pressure on the quantity and quality of the pre-collected meta data. Besides, these variables have different physical meanings and feature representations, and thus might better be optimized separately with specifically designed algorithms but not be entirely trained concatenatedly. Especially, if the training data or meta data contain certain amount of noises or outliers, such high non-convexity of the proposed model tend to make the them easily stuck to an unexpected local minimum. I thus think the initialization, especially for the NNs, should be extremely important to the final performance of the method. But this is always intractable in real applications. A suggestion might be pre-set a dictionary for generating NNS, with fixed pre-specified atoms, like those from training data by certain easy criterion. This should make the problem much easier and more stable. Another question is that when learning the NNs, the magnitude is neglected to be learned. But I think this magnitude should also be important for the final performance. I thus think clarifications on the rationality of such simplification. %%%%%%%%%%%% After reading the rebuttal provided by the authors and all discussions by reviewers, I still prefer to keep my original score. Thanks.


Review 3

Summary and Contributions: This paper proposes Meta-Neighborhoods that uses induced neighborhoods to summarize the training data into a much smaller dictionary. It further employs a meta-learning based training mechanism to jointly learn the induced neighborhood and the model. Extensive studies demonstrated the superiority of the proposed method.

Strengths: 1) The idea of meta-neighborhoods is novel and seems effective. 2) The paper performs extensive experiments to verify the effectiveness of the proposed method.

Weaknesses: 1) The author uses meta learning to learn the memory. However, there are some other works [1][2] uses a memory consisting of raw features. So What is the advantages of the meta-learning strategy to learn memory? 2) The ablation study lacks the experiments to verify the effectiveness of instance-wise FiLM models. 3) The Meta-Neighborhoods is semi-parametric, and is a generalizatio of k-nearest-neighbors. So what is parameters number and the time complexity of the proposed method compared to the baseline and k-nearest-neighbors, [1] Meta-learning with memory-augmented neural networks. ICML 2016. [2] Memory matching networks for one-shot image recognition. CVPR 2018. *******************After Rebuttal************************* I have carefully looked into all the reviews and rebuttals. The rebuttal addresses my concerns. I think the authors have performed significant experiments. It produces a good performance for large-scale tasks, 200-class classification on Tiny-Imagenet. It further reports the results on 1000-class ImageNet classification and also achieves significant gains. I agree with the other reviewers that the idea of combining single parameter initialization with non-parametric approaches for classification is interesting. So I will keep my score.

Correctness: Yes

Clarity: Yes. The method is clearly presented.

Relation to Prior Work: Yes. The paper discusses the relations of this work and related k-nearest-neighbors method.

Reproducibility: Yes

Additional Feedback: 1) The author uses meta learning to learn the memory. However, there are some other works [1][2] uses a memory consisting of raw features. So What is the advantages of the meta-learning strategy to learn memory? 2) The ablation study lacks the experiments to verify the effectiveness of instance-wise FiLM models? 3) The time complexity and parameter number of the proposed method? [1] Meta-learning with memory-augmented neural networks. ICML 2016. [2] Memory matching networks for one-shot image recognition. CVPR 2018.


Review 4

Summary and Contributions: This paper proposed to combine a discriminative classifier f and non-parametric neighbors to performance classification (or regression). The final classifier used for classification is obtained by fine-tuning f with neighbors of the testing example. The meta-neighbors (do not have to be training examples) are learnable. The contribution lies in: 1) Combine parametric and non-parametric methods in a meta-learning framework. 2) The experiment shows slightly better performance.

Strengths: 1. Combine parametric and non-parametric principles for classification is an interesting research direction. 2. The framework is simple and easy to reproduce.

Weaknesses: 1. Although having more information for classification during testing phase (meta neighbors), the experiment result did not show significant improvement. 2. One of the major draw back of a nonparametric method is that it does not produce good performance for large-scale classification tasks (lots of classes). The paper should apply the method to a larger dataset to study its behavior.

Correctness: yes

Clarity: yes

Relation to Prior Work: Meta-learning reference is very narrow presented. The paper should discuss a broader meta-learning literature.

Reproducibility: Yes

Additional Feedback:

[Author Response · NeurIPS 2020]

**Respond to Reviewer 1**   A common bias is that meta-learning should tackle transfer learning or few-shot learning problems. However, this is not always the case: *the setting of this paper **do not** fit nicely with transfer learning or few-shot learning*. This is because the learned neighbors are optimized using source domain data, which are useless and even harmful if we use them to adapt the model to unseen target domains. Similar to the setting of MAXL [1], the focus of our paper is to improve the general supervised learning performance via meta-learning.

As pointed out by ICLR 2019 AnonReviewer3 of the MAXL paper, "*Moreover, since the method is not a meta-learning approach for few-shot learning, it is not fair and also not appropriate to compare with Prototypical Network.*", we also think it is unreasonable to compare our work with MAML, prototypical networks and [2].

Table 1: Updated results for regression.

| Datasets | $n$ | $d$ | kNN | *vanilla* | Meta-Neighborhoods |
|---|---|---|---|---|---|
| music | 515345 | 90 | 0.6812±0.0062 | 0.6236±0.0056 | **0.6088±0.0050** |
| toms | 28179 | 96 | 0.0602±0.0083 | 0.0594±0.0080 | **0.0531±0.0073** |
| cte | 53500 | 384 | 0.00134±0.00023 | 0.00121±0.00022 | **0.00109±0.00015** |
| super | 21263 | 80 | 0.1126±0.0061 | 0.1132±0.0060 | **0.1077±0.0068** |
| gom | 1059 | 116 | 0.5982±0.0521 | 0.5949±0.0515 | **0.5681±0.0563** |

It is not advisable to evaluate the degree of improvements without considering the room available for improvements. Our improvements are **significant** as: (1) they are greater than those achieved by MAXL on almost all datasets (2) according to line 240-243, backbones used in our work are already strong, and our work is more effective than naively increasing the backbone depths. We report results on the 1000-class ImageNet classification. As shown in Fig 1, MN+iFiLM improve *vanilla* from 48.4% to 54.1%. Again, this improvement is larger than that achieved by MAXL. To facilitate experiments, we resize images to $64 \times 64$ resolution.

Figure 1: Top-1 Validation Accuracy on Imagenet.

For regression results, we provide results of kNN in Table 1, which are inferior to Meta-Neighborhoods. We also perform statistical significance test (paired Student's t-test) to show the results of Meta-Neighborhoods and *vanilla* are statistically different: the p-value of music, toms, cte, super and gom are 0.00039, 0.0018, 0.018, 0.0076 and 0.00089, which are all smaller than the Significance Level $\alpha = 0.05$.

We hope our response can address most of your concerns and sincerely hope you can re-consider your score.

**Respond to Reviewer 2**   In fact, we didn't observe optimization difficulties when training all variables together due to the following reasons: (1) we observed the pseudo-NNS can be easily initialized as Gaussian and not sensitive to the std of Gaussian (2) learning rate is only a scalar and thus easy to optimize (3) although the feature extractor receives error signals from the finetuned $\phi_i$, $\phi_i$ can be expressed as $\phi_i = \phi - \alpha \nabla_\phi L_i^{inner}$ where $\phi$ acts as a "short cut" to back-propagate errors to the feature extractor. Besides, our model is not sensitive to the choice of datasets. Neglecting magnitude actually does not harm the final performance as shown in [3]. On the contrary, it adds robustness by maximizing inter-class differences.

**Respond to Reviewer 3**   Both memory-augmented neural nets and memory matching nets tackle few-shot problems where the raw features are given, while our work does not consider few-shot tasks. Therefore, the raw features are not given in our case and we propose to meta-learn them. The effectiveness of iFiLM has been validated: MN+iFiLM is always better than MN. Please refer to Appendix A.3 and A.7 for parameter number and time complexity information.

**Respond to Reviewer 4**   It is not advisable to evaluate the degree of improvements without considering the room available for improvements. Our improvements are **significant** as: (1) they are greater than those achieved by current STOA method MAXL [1] on almost all datasets (2) backbones used in our work are already strong, which leaves limited room for large improvements. According to line 240-243, our work is more effective than naively increasing the backbone depths.

*Besides, our work has already produced a good performance for large-scale tasks that consist of many classes (e.g. 200-class classification on Tiny-Imagenet).* To validate this claim, we further report results on 1000-class ImageNet classification. As shown in Fig 1, MN+iFiLM improve *vanilla* from 48.4% to 54.1%. Again, this improvement is larger than that achieved by MAXL [1]. To facilitate experiments, we downsampled image resolution to $64 \times 64$.

Overall, we sincerely hope this response can address your concerns and you can re-consider your score.

**Reference**

[1] Self-supervised generalisation with meta auxiliary learning. NeurIPS 2019.

[2] Memory matching networks for one-shot image recognition. CVPR 2018.

[3] Robust classification with convolutional prototype learning. CVPR 2018


[Meta-Review · NeurIPS 2020]

The work presents a generalisation of k-nearest neighbours that compares favourably to recent work on wide range of supervised learning problems as demonstrated in the authors in the paper and their rebuttal. Some reviewers were concerned about the lack of comparison to MAML, but this was deemed unnecessary due to domain of experiments included.